

# Fisher linear discriminant analysis for classification and prediction of genomic susceptibility to stomach and colorectal cancers based on six STR loci in a northern Chinese Han population

Shuhong Hao[1], Ming Ren[2], Dong Li[3], Yujie Sui[4], Qingyu Wang[2], Gaoyang Chen[2], Zhaoyan Li[2] and Qiwei Yang[4]

[1] Department of Hematology and Oncology, The Second Hospital of Jilin University, Changchun, Jilin Province, China
[2] Department of Orthopedics, The Second Hospital of Jilin University, Changchun, Jilin Province, China
[3] Department of Obstetrics and Gynecology, The Second Hospital of Jilin University, Changchun, Jilin Province, China
[4] Medical Research Center, The Second Hospital of Jilin University, Changchun, Jilin Province, China

Corresponding author
Qiwei Yang, qiweiy@163.com

## ABSTRACT

**Objective**. Gastrointestinal cancer is the leading cause of cancer-related death worldwide. The aim of this study was to verify whether the genotype of six short tandem repeat (STR) loci including AR, Bat-25, D5S346, ER1, ER2, and FGA is associated with the risk of gastric cancer (GC) and colorectal cancer (CRC) and to develop a model that allows early diagnosis and prediction of inherited genomic susceptibility to GC and CRC.

**Methods**. Alleles of six STR loci were determined using the peripheral blood of six colon cancer patients, five rectal cancer patients, eight GC patients, and 30 healthy controls. Fisher linear discriminant analysis (FDA) was used to establish the discriminant formula to distinguish GC and CRC patients from healthy controls. Leave-one-out cross validation and receiver operating characteristic (ROC) curves were used to validate the accuracy of the formula. The relationship between the STR status and immunohistochemical (IHC) and tumor markers was analyzed using multiple correspondence analysis.

**Results**. D5S346 was confirmed as a GC- and CRC-related STR locus. For the first time, we established a discriminant formula on the basis of the six STR loci, which was used to estimate the risk coefficient of suffering from GC and CRC. The model was statistically significant (Wilks' lambda $= 0.471$, $\chi 2 = 30.488$, $df = 13$, and $p = 0.004$). The results of leave-one-out cross validation showed that the sensitivity of the formula was 73.7% and the specificity was 76.7%. The area under the ROC curve (AUC) was 0.926, with a sensitivity of 73.7% and a specificity of 93.3%. The STR status was shown to have a certain relationship with the expression of some IHC markers and the level of some tumor markers.

**Conclusions**. The results of this study complement clinical diagnostic criteria and present markers for early prediction of GC and CRC. This approach will aid in improving risk awareness of susceptible individuals and contribute to reducing the incidence of GC and CRC by prevention and early detection.

## INTRODUCTION

Gastrointestinal cancer is a leading cause of cancer-related death worldwide (*Zhou et al., 2017*). Of all types of gastrointestinal cancer, colorectal cancer (CRC) is the most common; CRC is the third most commonly diagnosed cancer in the United States and the fourth in China (*Du et al., 2017*; *Siegel et al., 2017*). Among CRC patients, 39% are diagnosed at the localized stage, for which the 5-year survival rate is 90%. The survival rate declines to 71% and 14% for patients diagnosed with regional and distant metastasis, respectively (*Siegel et al., 2017*). Although the incidence of gastric cancer (GC) is not as high as that of CRC in the United States, GC is the second most common malignancy in China (*Zheng et al., 2019*). Moreover, GC is the second most common cause of cancer-related death worldwide (*Torre et al., 2015*). Metastasis greatly affects the efficacy of treatment and is the main cause of fatality from the disease (*Gupta & Massagué, 2006*). Therefore, a deeper understanding of the molecular mechanisms in GC and CRC progression and the identification of early diagnostic biomarkers and predictive signals for the diseases have become urgent issues.

Short tandem repeats (STRs), also known as microsatellites, are small (1–6 base pairs) repeating DNA fragments occurring throughout the entire genome and account for approximately 3% of the human genome (*Nojadeh, Behrouz & Sakhinia, 2018*). These repeats are prone to accumulate mutations during DNA replication owing to a defect in the DNA mismatch repair (MMR) system (*Zeinalian et al., 2018*). Accumulating mutations in target genes including oncogenes, tumor suppressor genes, or cell cycle-regulating genes with microsatellite repeats lead to frameshift mutations, which can cause protein truncation and tumor progression (*Mokarram et al., 2014*). Several studies tested for somatic alterations in STR loci and determined that some of these mutations are associated with the diagnosis and prognosis of diseases including prostate (*Moya et al., 2018*), hepatocellular (*Rao, Asch & Yamada, 2017*), gastric (*Marrelli et al., 2016*), colorectal (*Yang et al., 2018*), breast (*McIver et al., 2014*), and lung (*Sozzi et al., 1999*) carcinomas. Most of these studies (*McIver et al., 2014*; *Marrelli et al., 2016*; *Rao, Asch & Yamada, 2017*; *Moya et al., 2018*; *Yang et al., 2018*) investigated microsatellite alterations, including microsatellite instability (MSI), and loss of heterozygosity (LOH) by analyzing the genotype in tumor tissues, with paired adjacent normal tissues as controls. It was also discovered that circulating cell-free tumor DNA (ctDNA) can be found in various cancers (*Bhangu et al., 2017*). Therefore, a small number of these studies (*Sozzi et al., 1999*) used ctDNA instead of tumor tissue DNA to analyze microsatellite alterations by comparison with normal lymph nodes. However, all these studies focused on the auto-microsatellite alterations in the patients themselves. To the best of our knowledge, there have been no reported studies involving comparison of STR loci between patients with cancer of the digestive tract and healthy controls to determine the risk coefficient of suffering from the diseases.

In previous studies, STR loci AR, BAT-25, D5S346, ER1, ER2, and FGA were reported to be closely related to the occurrence of malignant carcinomas. Androgen receptor (AR) is a nuclear transcription factor derived from the X-chromosome (*Ryan et al., 2017*). The AR gene contains a highly polymorphic repeat motif (CAG) in exon 1 encoding a glutamine tract that is inversely correlated with the transcriptional competence of the receptor (*Ackerman et al., 2012*). A previous study demonstrated that the CAG repeat sequence is closely related to the development of CRC and the long CAG repeat sequence is associated with a low 5-year overall survival (OS) rate in CRC patients (*Huang et al., 2015*). BAT-25, in introns of the c-kit oncogene, has been extensively studied for its major role in MSI (*Zhou et al., 1997*). Studies have revealed that MSI was detected in GC (*Polom et al., 2017*), CRC (*Esemuede et al., 2010*), and endometrial cancer (*Kanopiene et al., 2015*). D5S346 appears in introns of the tumor suppressor gene APC, which plays an important role in Wnt/β-catenin pathway regulation, cell migration, adhesion, chromosome segregation, and spindle assembly (*Zauber, Marotta & Sabbath-Solitare, 2016*). Another study demonstrated that LOH of the D5S346 locus occurred in more than 60% of esophageal carcinomas and almost all sporadic CRCs as well as many pancreatic and gastric carcinomas (*Nair, Naidoo & Chetty, 2006*). Estrogen receptors (ER) α and β (coded by the genes *ESR1* and *ESR2*, respectively), which are ligand-activated transcription factors, appear to play a predominant role in regulating cell proliferation and now serve as the basis for many therapeutic interventions (*Narita et al., 2010*). *Narita et al. (2010)* found that longer TA repeats in *ESR1* were correlated with increased disease-free survival in breast cancer patients. *Rudolph et al. (2014)* found that longer CA repeats in *ESR2* were potentially associated with an increase in CRC risk. FGA is a compound tetranucleotide repeat found in the third intron of the human alpha fibrinogen locus on the long arm of chromosome 4 (*Gettings et al., 2015*). A high incidence of alterations in FGA was observed in patients with GC (*Vauhkonen et al., 2004*) and oral cancer (*Pai et al., 2002*).

Therefore, in the present study, these six STR loci were analyzed using peripheral blood samples from six colon cancer patients, five rectal cancer patients, eight GC patients, and 30 healthy subjects with an automated fluorochrome-based DNA sequencer. At the same time, using the Fisher linear discriminant analysis (https://www.cnblogs.com/lyfruit/archive/2013/04/28/3048333.html) method, a new discriminant function used to estimate the risk coefficient of suffering from GC and CRC based on the allelic variations at polymorphic STR markers was established. Overall, this study provides further insight into the incidence and patterns of cancer of the digestive tract and will complement clinical diagnostic criteria for patients with these cancers. Whereas previous studies commonly focused on MSI or LOH in tumor tissues, herein we concentrated on genomic STR types, which are congenital rather than acquired, in the peripheral blood of GC and CRC patients and healthy individuals, allowing for the use of these STR loci in the prediction of inherited genomic susceptibility to GC and CRC in individuals.

**Table 1   Clinicopathological characteristics of the patients.**

| Factor | Gastrointestinal cancer ($n = 19$), $n$ (%) |
| --- | --- |
| Age (years) | |
| Mean $\pm$ SD | 64.7 $\pm$ 10.5 |
| Median | 64 (range, 46–88) |
| <60 | 6 (31.6) |
| ≥60 | 13 (68.4) |
| Sex | |
| Male | 13 (68.4) |
| Female | 6 (31.6) |
| Location | |
| Colon cancer | 6 (31.6) |
| Rectal cancer | 5 (26.3) |
| Gastric cancer | 8 (42.1) |
| Histological grade | |
| I | 0 (0.0) |
| II | 11 (57.9) |
| III | 5 (26.3) |
| IV | 3 (15.8) |
| Nodal status | |
| Positive | 11 (57.9) |
| Negative | 8 (42.1) |

## MATERIALS & METHODS

### Samples

A total of 19 samples from six colon cancer patients, five rectal cancer patients, and eight gastric cancer patients (as evidenced by pathological findings), recruited from The Second Hospital of Jilin University (Changchun, China) between December 2016 and December 2017, were included in the study (Table 1). In addition, 30 healthy subjects were recruited from the same city as the control groups. All participants were Han Chinese from northeast China.

### Ethics statement and informed statement

The Ethics Committee of the Second Hospital of Jilin University has a detailed understanding of and approved all experimental protocols in this study (approval number: 2016-39). This study conforms to the provisions of the Declaration of Helsinki (as revised in Fortaleza, Brazil, October 2013). All involved methods were carried out in accordance with relevant guidelines and regulations of the Ethics Committee of the Second Hospital of Jilin University. We informed all participants according to the informed consent for the use of their specimens, and written informed consents were obtained from each participant.

### DNA extraction and PCR amplification

Genomic DNA from 49 samples was extracted from 2 ml of peripheral blood using a Genomic DNA Kit (TIANGEN Biotech, Inc., Shanghai, China) according to the

**Table 2  Characteristics of each STR marker.**

| STR marker | Position | Start location | Repeat motifs | Gene |
|---|---|---|---|---|
| AR | Xq12 | 66657655 | CAG | *AR* |
| Bat-25 | 4q12 | 55633758 | T | *C-kit* |
| D5S346 | 5q21/22 | 111646983 | GT | *APC* |
| ER1 | 6q25 | 151806531 | TA | *ESR1* |
| ER2 | 14q23 | 64253561 | TG | *ESR2* |
| FGA | 4q31 | 154587748 | AAAG | *FGA* |

**Notes.**
Abbreviations: STR, short tandem repeat.

**Table 3  Primer sequences used for PCR amplification.**

| STR markers | Forward | Reverse | Dye |
|---|---|---|---|
| AR | AGGGCTGGGAAGGGTCTA | GGAGAACCATCCTCACCCT | HEX |
| Bat-25 | CGCCTCCAAGAATGTAAGTG | AACTCAAGTCTATGCTTCACCC | FAM |
| D5S346 | GGTTTCCATTGTAGCATCTTG | GCCTGGTTGTTTCCGTAGTA | FAM |
| ER1 | TCTGTTGGGTGTTTGGGATA | TTACATTGTCGGTCTGGTCC | ROX |
| ER2 | ATCTCAGTCTCCCCAAGTGC | TCCTTCAAGATAACCACCGA | FAM |
| FGA | TCGGTTGTAGGTATTATCACGG | TGCCCCATAGGTTTTGAACT | ROX |

**Notes.**
Abbreviations: HEX, 5′ hexachlorofluorescein; FAM, 5′ carboxyfluorescein; ROX, 5′ carboxy-X-rhodamine.

manufacturer's instructions. The extracted DNA was quantified using the NanoDrop 2000 spectrophotometer (Thermo Fisher Scientific, Waltham, MA, USA). Six (https://www.sciencedirect.com/topics/biochemistry-genetics-and-molecular-biology/microsatellite-dna) markers (AR, Bat-25, D5S346, ER1, ER2, and FGA) were amplified in the Mastercycler® nexus (Eppendorf, Hamburg, Germany) according to the manufacturer's instructions. The details of each STR marker are listed in Table 2. Each PCR reaction mixture contained 0.8 μl primer (the concentration of all primers was 10 μM), 30–300 ng template DNA, 10 μl PCR amplification reaction solution, and deionized water to a total volume of 20 μl. All primer sequences for amplification are listed in Table 3. The PCR reaction conditions were as follows: 95 °C for 3 min, followed by 10 cycles at 95 °C for 30 s, 60 °C for 30 s, and 72 °C for 30 s; 20 cycles of 95 °C for 30 s, 55 °C for 30 s, and 72 °C for 30 s; and a final extension step at 72 °C for 6 min.

## Fragment size determination

For determination of the size of each PCR product, the forward primer of each tested STR marker was labeled with a different fluorochrome: 5′ hexachlorofluorescein (HEX) for AR; 5′ carboxyfluorescein (FAM) for Bat-25, D5S346, and ER2; and 5′ carboxy-X-rhodamine (ROX) for ER1 and FGA (Table 3). Amplification products (1 μl) were mixed with 9.9 μl of formamide and 0.1 μl of an internal size standard (GenScan™ 500 LIZ® Size Standard; Applied Biosystems, Inc., Carlsbad, CA, USA). The mixture was denatured at 95 °C for 5 min and chilled for 5 min in an ice-water bath before capillary electrophoresis with a 3730XL Genetic Analyzer (Applied Biosystems, Inc.). The final size of the obtained

**Table 4** Formulas for calculating the copy number of each STR locus.

| STR markers | Copy number (X) |
| --- | --- |
| AR | X = round [(S-191)/3] |
| Bat-25 | X = round (S-379) |
| D5S346 | X = round [(S-202)/2] |
| ER1 | X = round [(S-359)/2] |
| ER2 | X = round [(S-278)/2] |
| FGA | X = round [(S-200)/4] |

**Notes.**
Abbreviations: X, the copy number of STR loci; S, the fragment length values of STR loci.

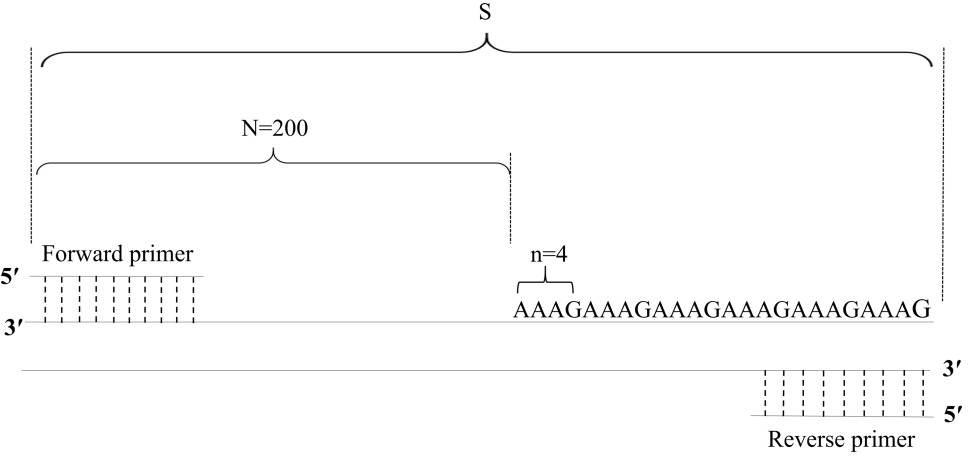

**Figure 1** Diagram of the method of calculation of copy numbers, with the FGA locus as an example.

PCR products was determined using the Gene Mapper ID software v3.2 (AB SCIEX, Framingham, MA, USA).

## STR typing determination

Based on the PCR product fragment sizes, the copy number of the STR loci was calculated with the following formula: X = round [(S–N)/n]. In this formula, X represents the copy number of the STR locus; S represents the fragment size of the PCR product; N represents the base number of the flanking region; n represents the base number of the repeat motifs; and round represents the omission of decimal fractions smaller than 0.5 and inclusion of all others, including 0.5, as 1 (Table 4, Fig. 1).

## Immunohistochemical (IHC) staining and assessment

All samples were fixed in 4% neutral formaldehyde solution and embedded in paraffin. Each tissue block was sectioned at 2-μm thickness, dewaxed, hydrated, and antigen-repaired by PT Link (Dako, Agilent Technologies, Santa Clara, CA, USA). Specifically, the sections were placed in the repair solution preheated to 65 °C, incubated for 30 min at 90 °C, and then cooled to 70 °C. Subsequently, the sections were washed with phosphate-buffered saline (PBS). Primary and secondary antibodies and 3,3′-diaminobenzidine (DAB) solution

were applied with the Autostainer Link 48 (Dako, Agilent Technologies). In brief, the sections were incubated with hydrogen peroxide for 10 min, primary antibody for 30 min, and secondary antibody for 20 min at room temperature. Finally, the sections were counterstained with hematoxylin, routinely dehydrated, cleared, and sealed.

All cases were divided into two groups. EGFR and CDX2 protein expression was determined based on the detection of staining at a particular location: –(no staining) and + (staining). Ki67 and P53 protein expression was determined based on the percentage of immunoreactive cells, which was defined as low expression (<50%) and high expression (≥50%).

## Tumor marker detection

Serum tumor markers were measured by chemiluminescence immunoassay using Quantitative Assay Kits (Tegen, Shanghai, China). The normal cutoff values for these markers were as follows: CA199 <35 U/ml, CEA <5 ng/ml, CA125 <35 U/ml, CA242 <20 U/ml, and CA72-4 <6.9 U/ml.

## Statistical analysis

Statistical analysis was performed using the SPSS Statistics 22.0 software (SPSS Inc., Chicago, IL, USA). Normally distributed variables were compared by Student's $t$-test and presented as means ± standard deviation. According to the STR locus typing, FDA was performed to establish the linear combination to distinguish the GC group from the healthy group. FDA served as a dimension reduction tool to identify a linear function of the variables to maximize differences between samples of multiple classes and to minimize differences between samples of the same class (*Zou et al., 2018*). Leave-one-out cross validation, in which each case was classified according to the formula derived from all cases other than that case, was used to validate the accuracy of the FDA. Receiver operating characteristic (ROC) curves were also used to validate the accuracy of the FDA; an area under the ROC curve (AUC) between 0.5 and 0.7 represented low diagnostic value, between 0.7 and 0.9 represented medium diagnostic value, and more than 0.9 represented high diagnostic value. The results were considered statistically significant at $p < 0.05$. The correspondence between the STR locus typing and IHC and tumor markers was analyzed using multiple correspondence analysis, which is a multivariate statistical technique used to explore the simultaneous relationships between variables (*Leandro-Merhi & JLB, 2017*).

## RESULTS

### Fragment size determination

The fragment size of each STR locus in all samples was detected (Dataset S1). An example electropherogram is shown in Fig. 2.

### STR typing determination

We used the copy number of each STR locus to represent the STR typing. The calculation of the copy number is detailed in Table 4, the results of which are shown in Fig. 3 and Dataset S2. Because each STR locus contains two alleles, whose copy numbers may be

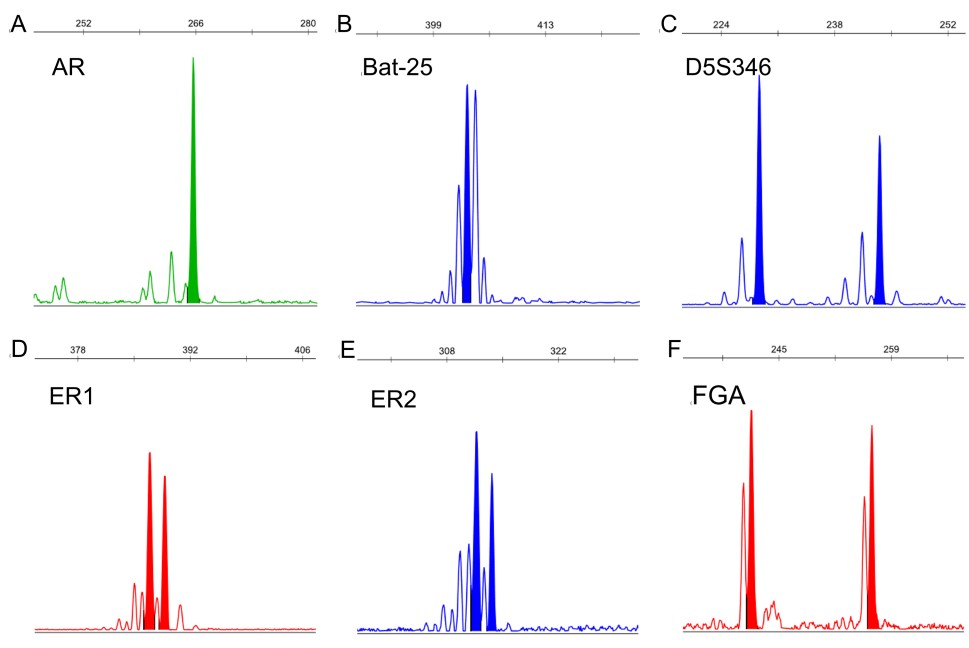

**Figure 2** Example electropherograms of AR (A), Bat-25 (B), D5S346 (C), ER1 (D), ER2 (E), and FGA (F). The numbers above each peak indicate the allele fragment length. Electropherograms containing one peak (A and B) represent homozygosity, whereas electropherograms containing two peaks (C–F) represent heterozygosity.

different, we divided each STR locus into two indexes: short repeat motif (marked as "-S") and long repeat motif (marked as "-L"). By comparing the copy number of each STR locus between GC and CRC patients and healthy subjects, we found that the copy number of AR-L was lower in GC and CRC patients than in healthy subjects ($26.16 \pm 2.91$ vs. $27.80 \pm 2.17$, $p = 0.032$); the copy number of D5S346-S was higher in GC and CRC patients than in healthy subjects ($14.84 \pm 2.66$ vs. $13.23 \pm 1.45$, $p = 0.027$); the copy number of D5S346-L was higher in GC and CRC patients than in healthy subjects ($18.79 \pm 4.21$ vs. $15.50 \pm 3.95$, $p = 0.009$); and the copy number of ER2-L was lower in GC and CRC patients than in healthy subjects ($19.74 \pm 2.77$ vs. $21.30 \pm 2.19$, $p = 0.037$). There was no significant difference in the copy number of other STR loci between the GC and CRC patients and healthy subjects ($p > 0.05$).

## Establishment of discriminant function

To distinguish the GC and CRC group from the healthy group based on the copy number of STR loci, an FDA of the 12 alleles of the six STR loci was performed. According to the results of statistical analysis, we obtained the following discriminant functions (Wilks' lambda $= 0.471$, $\chi^2 = 30.488$, $df = 13$, and $p = 0.004$):

$Cfunc_1$ = 12.431 X1–3.900 X2+16.475 X3+117.320 X4–2.841 X5–0.399 X6+0.971 X7+7.972 X8+0.185 X9–3.214 X10+6.034 X11–3.737 X12–28.835 X13–1776.227 (GC patient);

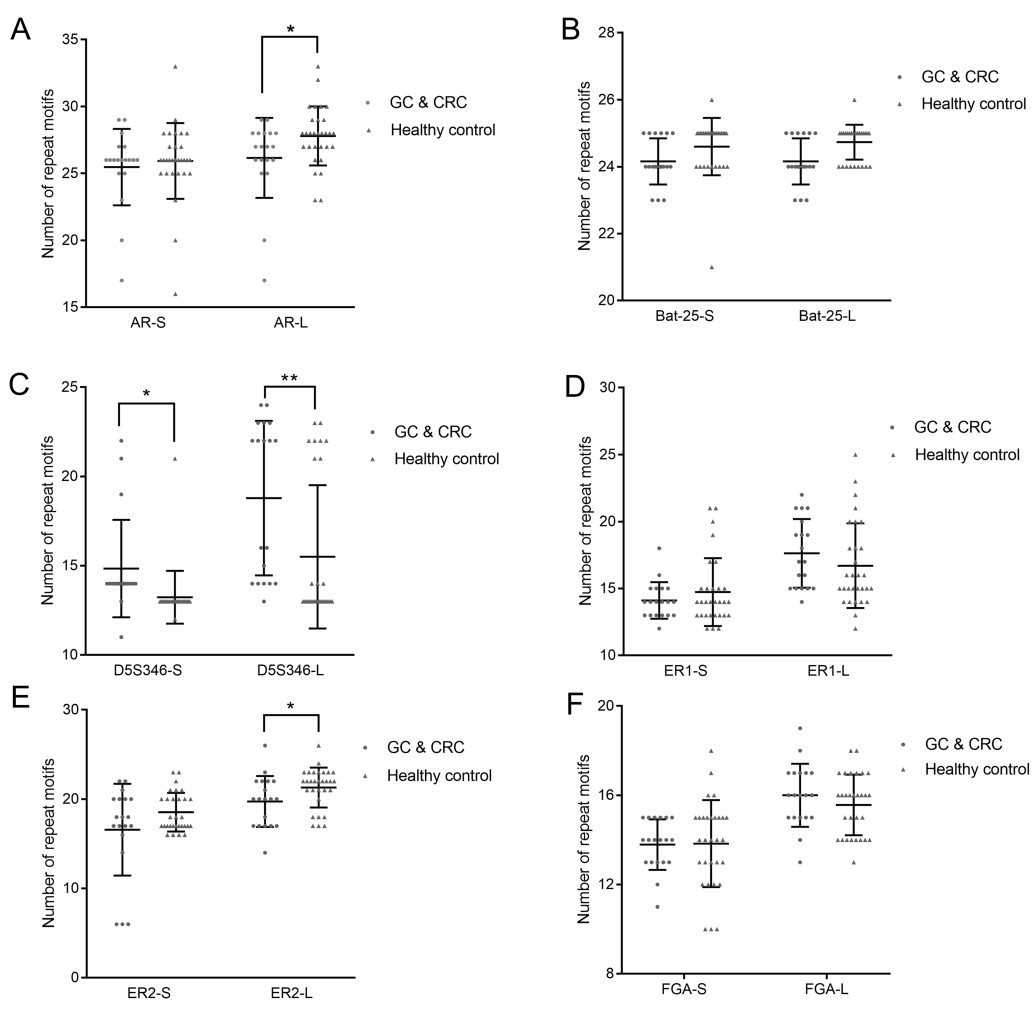

**Figure 3  Copy number of AR (A), Bat-25 (B), D5S346 (C), ER1 (D), ER2 (E), and FGA (F) between GC and CRC patients and healthy controls.** "-S" represents the short repeat motif of STR alleles; "-L" represents the long repeat motif of STR alleles. GC, gastric cancer; CRC, colorectal cancer. Statistical significance is indicated by asterisks: *, $p < 0.05$; **, $p < 0.01$.

$\text{Cfunc}_2 = 12.290\ X1 - 3.378\ X2 + 16.342\ X3 + 118.680\ X4 - 3.252\ X5 - 0.757\ X6 + 1.308\ X7 + 7.836\ X8 + 0.163\ X9 - 2.917\ X10 + 5.877\ X11 - 4.148\ X12 - 30.834\ X13 - 1803.632$ (healthy control).

In the discriminant formulas, X1–X12 represent the copy number of different STR loci (Table 5). If the subject is female, X13 was zero; if the subject is male, X13 was one. The values of $\text{Cfunc}_1$ and $\text{Cfunc}_2$ were calculated by substituting the copy number of STR loci into the discriminant function. By comparing the $\text{Cfunc}_1$ and $\text{Cfunc}_2$ values, subjects with unknown classification were classified according to the following principles: if $\text{Cfunc}_1 > \text{Cfunc}_2$, the subjects were classified into the GC and CRC group; if $\text{Cfunc}_1 < \text{Cfunc}_2$, the subjects were classified into the healthy group.

**Table 5    The STR loci corresponding to X1–X12 in the discriminant formula.**

| Variables | STR loci |
|---|---|
| X1 | AR-S |
| X2 | AR-L |
| X3 | Bat-25-S |
| X4 | Bat-25-L |
| X5 | D5S346-S |
| X6 | D5S346-L |
| X7 | ER1-S |
| X8 | ER1-L |
| X9 | ER2-S |
| X10 | ER2-L |
| X11 | FGA-S |
| X12 | FGA-L |

**Table 6    Result of cross-validation.**

| Actual | Predicted ($n$, %) | |
|---|---|---|
| | **GC and CRC patients** | **Healthy controls** |
| GC and CRC patients | 14(73.7) | 5(26.3) |
| Healthy controls | 7(23.3) | 23(76.7) |

**Notes.**

Abbreviations: GC, gastric cancer; CRC, colorectal cancer.

## Accuracy and diagnostic value of the discriminant formula

Leave-one-out cross validation was performed to validate the accuracy of the discriminant formula. The results showed that in the GC and CRC group, 14 patients (73.7%) were correctly classified, with a misclassification rate of 26.3%; in the healthy group, 23 cases (76.7%) were correctly classified, with a misclassification rate of 23.3% (Table 6).

An important measure to validate the accuracy of a prediction formula is the ROC curve.

In the present study, ROC curves were generated based on the value of Cfunc (Cfunc$_1$–Cfunc$_2$) calculated for each case (range from –5.896 to 5.434) and the copy number of all the STR loci. The results showed that the area under the ROC curve (AUC) of Cfunc was 0.926 (95% CI [0.851–1.000]), which indicated a high diagnostic value (Fig. 4). Considering 0 as the criterion for differential diagnosis, the sensitivity of the ROC curve was 0.737, and the specificity was 0.933. Among the STR loci, the AUC of D5S346-S was 0.894, and the AUC of D5S346-L was 0.798, which indicated a medium diagnostic value. Contrarily, the AUC of other STR loci was between 0.305 and 0.622, which indicated no or low diagnostic value (Fig. S1, Table 7).

## Correspondence between STR typing and IHC markers

In order to explore the correlation between the STR locus status and commonly used IHC markers, including EGFR, CDX2, Ki67, and P53, Student's $t$-test and multiple correspondence analysis were performed. The results of Student's $t$-test showed that there

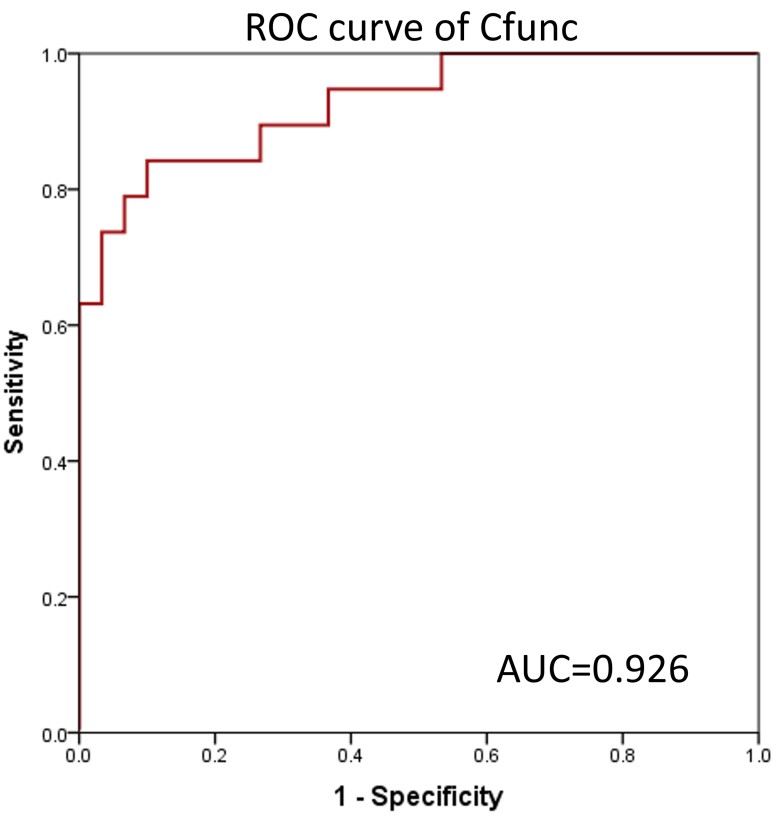

**Figure 4** ROC curve of Cfunc.

**Table 7** Coordinates of ROC curve for Cfunc and copy number of all STR loci.

| Method | AUC | P | 95% CI | |
|---|---|---|---|---|
| | | | Upper limits | Lower limits |
| Cfunc | 0.926 | 0.000 | 0.851 | 1.000 |
| AR-S | 0.479 | 0.806 | 0.312 | 0.646 |
| AR-L | 0.332 | 0.050 | 0.180 | 0.484 |
| Bat-25-S | 0.454 | 0.587 | 0.288 | 0.619 |
| Bat-25-L | 0.430 | 0.412 | 0.264 | 0.595 |
| D5S346-S | 0.894 | 0.000 | 0.771 | 1.000 |
| D5S346-L | 0.798 | 0.000 | 0.674 | 0.923 |
| ER1-S | 0.473 | 0.750 | 0.310 | 0.635 |
| ER1-L | 0.622 | 0.154 | 0.465 | 0.779 |
| ER2-S | 0.446 | 0.525 | 0.274 | 0.618 |
| ER2-L | 0.305 | 0.023 | 0.150 | 0.461 |
| FGA-S | 0.476 | 0.782 | 0.315 | 0.637 |
| FGA-L | 0.582 | 0.335 | 0.419 | 0.746 |

**Notes.**
Abbreviations: ROC, receiver operating characteristic; AUC, area under the ROC curve; STR, short tandem repeat.

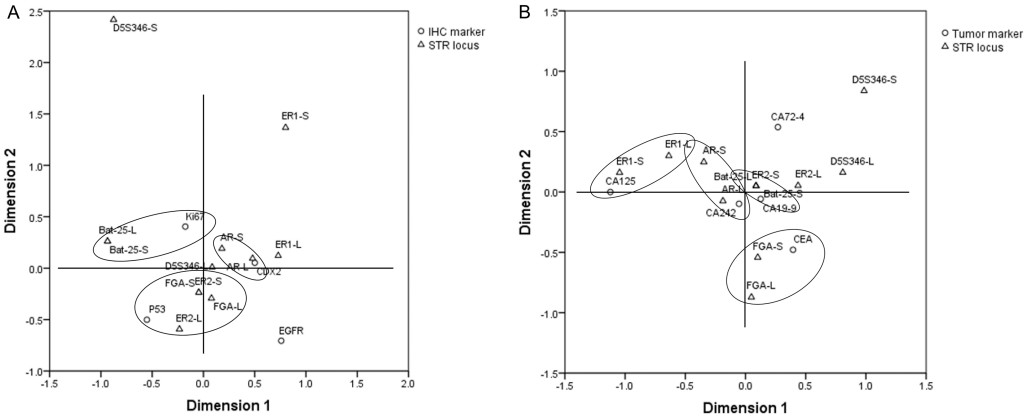

**Figure 5** **Results of multiple correspondence analysis.** (A) The correlation between the copy numbers of STR loci and IHC markers in GC and CRC. (B) The correlation between the copy numbers of STR loci and tumor markers in GC and CRC. GC, gastric cancer; CRC, colorectal cancer.

was no significant difference in the copy number of STR loci between the IHC (+)/high expression group and IHC (−)/low expression group ($p > 0.05$) (Fig. S2). The results of multiple correspondence analysis showed that the copy number of AR was related to the expression of CDX2; the copy number of Bat-25 was related to the expression of Ki67; and the copy numbers of ER2 and FGA were related to the expression of P53 (Fig. 5A).

## Correspondence between STR typing and tumor markers

Student's $t$-test and multiple correspondence analysis were performed to explore the correlation between the STR locus status and commonly used tumor markers, including CA125, CA19-9, CA242, CA72-4, and CEA. The results of Student's $t$-test showed that there was no significant difference in the copy number of STR loci between the normal and abnormal tumor marker groups ($p > 0.05$) (Fig. S3). The results of multiple correspondence analysis showed that the copy number of AR was related to the level of CA242; the copy number of Bat-25 was related to the level of CA19-9; the copy number of ER1 was related to the level of CA125; and the copy number of FGA was related to the level of CEA (Fig. 5B).

## DISCUSSION

It is widely accepted that cancer is influenced by many genes that interact as well as the environment (*Qi et al., 2018*). Therefore, single genetic alterations do not provide sufficient information about a given disease. In the present study, we established a new method that can be used for early diagnosis and prediction of genomic susceptibility to GC and CRC in a Han population in northern China by integrated analysis of multiple STR loci using FDA.

Initially, on the basis of several previous studies, we focused on six different STR loci, including AR, BAT-25, D5S346, ER1, ER2, and FGA, which were reported to be closely related to the occurrence of malignant carcinomas. We divided each STR locus into two allelotypes, short repeat motif and long repeat motif. This method avoids influence from

two alleles at one STR locus with different numbers of copies; each allele has the same average copy number compared with that of another individual. By comparing the copy number of a single STR locus between GC and CRC patients and healthy subjects, we found that the copy number of both D5S346-S and D5S346-L was higher in GC and CRC patients than in healthy subjects ($p < 0.05$ and $p < 0.01$, respectively). This result indicated that D5S346 is a GC- and CRC-related STR locus, which is similar to results of previous studies (*Stemmermann et al., 1994*; *Nair, Naidoo & Chetty, 2006*). Then, we compiled these 12 alleles to conduct an integrated analysis using the FDA method. FDA is a classical approach to identify a linear function of variables to distinguish samples from different groups as much as possible (*Zou et al., 2018*). This method has been widely applied in many fields, including facial recognition (*Ruiz-Del-Solar & Navarrete, 2005*), market research (*IMP et al., 2018*), and disease classification (*Zou et al., 2018*). In the present study, we successfully established a discriminant formula to distinguish patients with GC and CRC from healthy controls using the FDA method ($p < 0.01$). The result of the leave-one-out cross validation revealed that the sensitivity of the discriminant formula was 0.737 and the specificity was 0.767. We also used another method, ROC curve analysis, to validate the accuracy of the discriminant formula based on the value of Cfunc, which was calculated by substituting the copy number of each STR locus into the function. The results showed that the AUC of Cfunc was 0.926, which was higher than the AUC of D5S346 (0.894 and 0.798) and other STR loci (0.300–0.610). This result indicated that the diagnostic and predictive value of the discriminant formula was higher than that of the single D5S346 locus, while other STR loci had no significant diagnostic and predictive value. In addition to gastrointestinal cancer, we have conducted similar studies in other cancers, such as lung cancer and breast cancer. Some results of the previous studies are similar to the current findings. However, the potentially valuable STR loci identified in the other cancers did not completely coincide with the six loci identified in the current study.

Although STR alterations have been widely studied in GC and CRC, no studies to our knowledge have compared these polymorphic genotypes with IHC and tumor markers. In this context, we intended to study the relationships between them using Student's *t*-test and multiple correspondence analysis. Four IHC markers (EGFR, CDX2, Ki67, and P53) were selected for this study, all of which were previously reported to be associated with the occurrence, diagnosis, or prognosis of GC or CRC. Epidermal growth factor receptor (EGFR) is a transmembrane glycoprotein that represents one of the four members of the ErbB family of tyrosine kinase receptors (*Singh et al., 2016*). Overexpression of EGFR is frequently associated with metastasis and therapy of CRC (*Yazdi et al., 2015*). Caudal-related homeobox transcription factor (CDX2) is an intestinal transcription factor whose loss is associated with high tumor grade, advanced stage in CRC (*Reggiani et al., 2017*), and poor prognosis in GC (*Wang et al., 2012*). Ki67 is a nuclear proliferation-associated antigen that is expressed in the growth and synthesis phases of the cell cycle but not in the resting phase. The Ki67 proliferation index increases during the transformation of intestinal metaplasia to gastric carcinoma (*Zheng et al., 2010*). P53 is encoded by *TP53* gene, which is one of the most important tumor suppressor genes and the main cell-cycle checkpoint. The high expression of dysfunctional P53 is common in CRC, as it plays a

role in the classical adenoma to carcinoma succession (*Mármol et al., 2017*). In the present study, because of the limited sample size, Ki67 and P53 protein expression was divided into only two groups (low expression and high expression groups) instead of smaller subgroups, unlike the clinical method used to score these markers. The results of Student's *t*-test indicated that there was no relationship between STR loci and IHC or tumor markers. However, the results of multiple correspondence analysis indicated that the copy number of AR was related to the expression of CDX2 and the level of CA242; the copy number of Bat-25 was related to the expression of Ki67 and the level of CA19-9; the copy number of ER1 was related to the level of CA125; the copy numbers of ER2 and FGA were related to the expression of P53; and the copy number of FGA was related to the expression of P53 and the level of CEA. The specific mechanism and clinical significance of the relationship between STR loci and IHC or tumor markers require further experiments for confirmation.

There are some limitations in the present study. Because of the insufficient sample size, we did not perform correlation analysis between STR variation and patient clinical and pathological characteristics. We believe that more valuable results, which may provide new insight into prognostic evaluation and treatment determination, will be obtained with a larger sample size.

## CONCLUSIONS

In conclusion, we confirmed that D5S346 is a GC- and CRC-related STR locus. For the first time, we established a discriminant formula, which was used to estimate the risk coefficient of GC and CRC in a Han population residing in northern China as well as predict inherited genomic susceptibility to the diseases by analyzing the STR typing of genomic DNA. These findings provide supporting evidence for use of STR markers as diagnostic and susceptibility biomarkers for GC and CRC patients. Additionally, we revealed that a relationship between the STR locus and IHC or tumor markers may exist. Nonetheless, the specific mechanism and clinical significance of the relationship require further experiments for confirmation.

### Funding

This work was supported by grants from the Education Department of Jilin Province, P.R.C. (No. JJKH20190059KJ), the Science and Technology Department of Jilin Province, P.R.C. (No. 20180520111JH), the Science and Technology Department of Jilin Province, P.R.C. (No. 20190201050JC), and the Second Hospital of Jilin University, P.R.C. (KYPY2018-02). The funders had no role in study design, data collection and analysis, decision to publish, or preparation of the manuscript.

### Grant Disclosures

The following grant information was disclosed by the authors:
Education Department of Jilin Province, P.R.C: No. JJKH20190059KJ.
Science and Technology Department of Jilin Province, P.R.C: No. 20180520111JH, No. 20190201050JC.

Second Hospital of Jilin University, P.R.C: KYPY2018-02.

## Competing Interests

The authors declare there are no competing interests.

## Author Contributions

- Shuhong Hao performed the experiments, authored or reviewed drafts of the paper.
- Ming Ren performed the experiments.
- Dong Li and Gaoyang Chen prepared figures and/or tables.
- Yujie Sui and Qingyu Wang analyzed the data.
- Zhaoyan Li contributed reagents/materials/analysis tools.
- Qiwei Yang conceived and designed the experiments, authored or reviewed drafts of the paper, approved the final draft.

## Human Ethics

The following information was supplied relating to ethical approvals (i.e., approving body and any reference numbers):

The study was approved by the Ethics Committee of The Second Affiliated Hospital of Jilin University (Ethical Application Ref: 2016-39).

## Data Availability

The raw measurements are available in Datasets S1 and S2.

## Supplemental Information

Supplemental information for this article can be found online at http://dx.doi.org/10.7717/peerj.7004#supplemental-information.

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
