# Peer review of "Fisher linear discriminant analysis for classification and prediction of genomic susceptibility to stomach and colorectal cancers based on six STR loci in a northern Chinese Han population"

_PeerJ, doi:10.7717/peerj.7004_

## Round 0.1 · original submission · Major Revisions

Both reviewers agree that this is a well written and well designed study, but that the manuscript needs more explanation regarding selection of the STR markers. Please consider performing additional analyses to address the heterogeneity of the cancers under study - such as the addition of more GCs. Moreover, as the sample size is small and restricted to one ethnicity, these caveats ought to be addressed fully in the discussion, and the translational potential of the findings should be presented with more cautious language.

·

Basic reporting

Clear professional English language is used throughout the manuscript. Both Introduction and Discussion provide the appropriate context for the study. References are relevant and up to date. Tables and Figures contain pertinent data, are well written and formatted. The authors are advised to detail what guided the choice of immunohistochemical markers tested in this study. Considering the large plethora of immunohistochemical markers, why were the ones included here prioritized versus others available?

Experimental design

The study is well designed and asks an original and meaningful question. The molecular methods are described in sufficient detail to be replicated. The investigation appears to abide to high technical standards.
The authors are requested to explain the assigned cut-offs for ki-67 and p53 immunostains. In clinical practice ki-67 is scored as an approximate percentage of tumor cell nuclei positive and p53 is interpreted as representing wild-type or aberrant expression. The cut-offs of >50% or <50% assigned by the authors may not be relevant.
It would be highly valuable to do MSI analysis of the tumor tissue in patients in the study group and the authors are advised to do so if possible.
No ethical standards for the study have been included. The authors are asked to specify if the subjects have consented to participate in the study and if institutional approval has been obtained.

Validity of the findings

The validity of the findings is limited by the small number of subjects. In addition, the study appears to include only the Chinese ethnic group and may not be representative of other populations. As only gastrointestinal cancer and controls were studied, it is not possible to ascertain if the STR loci implicated are specific for gastrointestinal cancer or might be associated with other cancer types.

Additional comments

Well-written and technically sound study with interesting findings. The molecular data is the most relevant. The immunohistochemical and serum marker data have not been well developed in this work and does not appear to add much to the value of the publication.

·

Basic reporting

This is an overall relatively well written, structured and cited manuscript. However, there are a few areas that could use improvement.

1) I feel that the introduction does not adequately explain why the six STR loci were chosen. Luckily the authors included this information in their discussion (lines 258-287) and I believe that this would be much more appropriate in the introduction.

2) The opening lines of the introduction (lines 63-72) are a little confusing as gastrointestinal (GI) carcinoma as a group is not defined. Additionally, the reference supplied says that GI carcinoma is A leading cause of cancer related deaths not THE leading cause and I think that is an important distinction to make in line 63. Some reference to the geographic and ethnic discrepancies in incidence would also be appropriate.

3) Line 84 should reference that the subsequent studies all tested for somatic alterations in STR loci.

4) The data presented in the "Establishment of discriminant function" section (lines 206-222) is quite difficult to read in its current form. Construction of one or more tables would help immensely.

5) Table 1 column two header currently has "Gastrointestinal cancer (n=30), n (%)" should be "... (n=19) ..."

6) Figure 4 is rather cluttered in my opinion and would better off split into two figures (one with the cfunc in the main paper, and the other with the loci in a supplemental table)

Experimental design

The question asked by this manuscript is quite interesting and worth exploring. However, I believe that the current structure of analysis is not fully able to answer it.

The authors combine 19 colon, rectal and gastric carcinomas and compare this against normal controls. While colon and rectal carcinomas are generally considered relatively comparable, gastric carcinomas are different enough (both clinically and genomically) that it is inappropriate in my opinion to group them together. Additionally, GI carcinomas also include pancreatobilliary, esophageal, small intestinal and hepatic carcinomas, none of which are represented in this manuscript. I believe that this article would be significantly stronger if the authors analyzed additional cases and separated out gastric from colorectal into separate analyses. If this was done a revision to "select gastrointestinal carcinomas" throughout the article would also be appropriate.

Validity of the findings

In the "Correspondence between STR typing and IHC markers" and "... and tumor markers" sections (lines 238-249) no statistical analysis is provided to support the claims of relation between the STR typing and either IHC or serum tumor markers. This should be performed and reported as such.

---

## Round 0.2 · Minor Revisions

The comment by reviewer 1 relating to p53 and ki67 cutoffs is well taken, but the study size precludes breaking down the cohort into smaller subgroups. I suggest that the authors mention that their approach is different to the clinical method of scoring these markers as a caveat in the discussion. The paper should then be acceptable for publication.

·

Basic reporting

Explaining the choice of IHC markers is a relevant addition to the manuscript (lines 332-346)

Experimental design

The authors choose to keep the 50% cut-off for Ki-67 and p53. This is acceptable albeit far from ideal for Ki-67.
It does not reflect tumor biology for p53. Advise the authors to either remove p53 from the paper or to score it was wild-type or mutant expression pattern.
Ethical parameters were appropriately included.

Validity of the findings

Authors have clarified the limitation of sample size and restricted ethnic subgroup included in the study.
Authors are advised to include in the Discussion the data provided to the reviewers: 'In addition to gastrointestinal cancer, we have conducted similar studies in other cancers, such as lung cancer and breast cancer. Some results are similar to the current findings, but not exactly the same. Potentially valuable STR loci found in other cancers are not exactly matched by these six loci.'

Additional comments

In my view, the serum and immunohistochemical markers are of little if any relevance for this study. I would be in favor of removing this data from this manuscript. Authors may consider further studies and submission of a separate manuscript correlating to serum and IHC makers to this or some other journal with more solid data.

·

Basic reporting

The reporting is much clearer with this revision. Great work!

Experimental design

I still believe that this will be a stronger study with more samples, but I accept the limitations that your institution has. The disclaimers at the end of the paper i think adequately inform readers that this is still a pilot study. I look forward to any follow up studies!

Validity of the findings

Edits addressed this section.

---

## Round 0.3 · accepted · Accept

Thank you for the edits to the discussion, the report is now suitable for publication.

#